# Nursing and midwifery students' attitudes towards addressing patient sexual health in their future profession: Polish adaptation and validation of the students' attitudes towards addressing sexual health extended questionnaire (SA-SH-Ext)

**Barbara Ślusarska[1], Ludmiła Marcinowicz**[2]*

**1** Department of Family and Geriatric Nursing, Chair of Integrated Nursing Care, Faculty of Health Sciences, Medical University of Lublin, Lublin, Poland, **2** Department of Developmental Period Medicine and Paediatric Nursing, Medical University of Bialystok, Bialystok, Poland

* ludmila.marcinowicz@umb.edu.pl

**Citation:** Ślusarska B, Marcinowicz L (2024) Nursing and midwifery students' attitudes towards addressing patient sexual health in their future profession: Polish adaptation and validation of the students' attitudes towards addressing sexual health extended questionnaire (SA-SH-Ext). PLoS ONE 19(6): e0300515. https://doi.org/10.1371/journal.pone.0300515

**Data Availability Statement:** All relevant data are within the paper and its Supporting information file.

## Abstract

The aim of the study was to assess the success of an adaption of the Students' Attitudes Towards Addressing Sexual Health Extended Questionnaire (SA-SH-Ext) in meeting Polish linguistic and cultural norms, as well as to ascertain the nursing and midwifery students' attitudes towards addressing sexual health using the SA-SH-Ext questionnaire. The sample size of the cross-sectional validation study consisted of 570 Polish nursing and midwifery students. The collected data was used to examine the internal consistency reliability and construct validity using exploratory factor analysis (EFA). Internal consistency reliability showed a Cronbach's alpha value of 0.91, and construct validity measured by exploratory factor analysis (EFA) demonstrated good results. The Kaiser-Meyer-Olkin measure of sampling adequacy (KMO) was high and amounted to 0.923, and the Bartlett's test of sphericity was significant (p = 0.000). The analysis of construct validity demonstrated five major factors: "Present feelings of comfortableness" (Factor 1), "Future working environment" (Factor 2), "Fear of negative influence on future patient relation" (Factor 3), "Educational needs—Awareness of knowledge gap" (Factor 4), "Educational needs—Awareness of the needs for competences" (Factor 5). The SA-SH-Ext v.PL questionnaire is a reliable and valuable instrument for assessing the level of perceived preparedness among nursing and midwifery students in addressing patient sexual health, a field often neglected in health and holistic care.

**Funding:** The author(s) received no specific funding for this work.

**Competing interests:** The authors have declared that no competing interests exist.

## Introduction

Sexual health is fundamental to the overall health and well-being of individuals and families, and to the social and economic development of communities and nations. According to the World Health Organization (WHO), sexual health is an essential element of human overall health throughout life and it is defined as "*a state of physical, emotional, mental and social well-being in relation to sexuality; it is not merely the absence of disease, dysfunction or infirmity*" [1]. In clinical and health care practice, health care professionals must protect, respect, and meet their patients' basic health needs, including sexual health well-being [2, 3]. However, research shows that sexual health care is often ignored in professional practice [4–6]. According to a holistic perspective of the healthcare facility, the profession of a nurse or midwife involves professional diagnostic and care activities in all areas of human health. As a result, a certain degree of education is required, including in the sexual health-related area. However, since sexual health issues are regarded as highly personal, delicate and sensitive, the specificity of knowledge and skills in this area is frequently disregarded in professional education and nursing care. Many nurses and midwives fail to have crucial discussions regarding sexual health with patients, which can cause problems in maintaining well-being and may even lead to adverse health occurrences [4, 7, 8]. The literature review shows that there are several significant barriers that medical professionals, such as nurses and midwives, face when trying to provide sexual health care. These include a lack of preparation during the professional education stage, insufficient knowledge and skills, a lack of professional training, feelings of embarrassment among health professionals, and health professionals questioning whether sexual health is part of their professional responsibility [3, 5, 7, 9, 10]. In addition to professional education, cultural and social factors also have an impact on attitudes towards sexual health. In order to effectively care for patients' sexual health in their future profession, students in the health care and social fields should be equipped with sufficient knowledge to overcome obstacles such as socio-cultural sensitivity to sexual matters, personal beliefs, staff narratives, and negative attitudes from colleagues [11].

The sexual health and care competencies in the professional education curriculum for health care professionals varies amongst countries. Notably, the extent of learning outcomes constituting the basis for the required competences for sexual health care is not clearly distinguished in Poland's professional education system for the professions of nurse and midwife, as per the applicable education standards for practising them [12]. It must be recognized that certain elements are implemented in school medicine to prepare students for the nursing profession [13], and for midwifery students in the module of family planning, motherhood and fatherhood [14]. However, the findings of a Polish study conducted among medical university students reveal knowledge gaps regarding a variety of sexuality-related aspects [15, 16]. Numerous past studies conducted in other countries also show that medical students acknowledge the need for education to help them deal with sexual health and sexuality with regard to interacting with patients, but they also admit that they feel unprepared to do so [17–21].

Education on sexual health varies within the professional education curriculum for nurses and midwives in Poland, which may indicate that there is a risk of varying levels of preparedness to care for sexual health in their future profession. As a result, it is necessary to assess students' competences and preparedness to provide sexual health care in the country. Such research has not yet been conducted in Poland. In addition, it is important to understand how Polish nursing and midwifery students view issues related to sexual health. It can also be presumed that students have insufficient knowledge and are not prepared to care for sexual health, and students' perceived competence and preparation may differ depending on their field of study. It is, therefore, important to evaluate Polish nursing and midwifery students'

present attitudes, perceived competencies and educational needs in this area in order to determine how best to instruct them about sexual health care in future professional interventions.

The aim of the study was to assess the success of an adaptation of "The Students' Attitudes Towards Addressing Sexual Health Extended Questionnaire (SA-SH-Ext)" in meeting Polish linguistic and cultural specifications, as well as to determine the nursing and midwifery students' attitudes towards addressing sexual health using the modified SA-SH-Ext questionnaire.

## Material and methods

### Design

This is a cross-sectional validation study conducted in accordance with the Strengthening the Reporting of Observational Studies in Epidemiology guidelines (STROBE) [22]. The psychometric assessment of The Students´ Attitudes Towards Addressing Sexual Health Extended Questionnaire (SA-SH-Ext) was conducted on a sample of nursing and midwifery students in Poland.

### Sample

In our research, a convenience sampling method was used. All undergraduate and graduate students enrolled full-time in nursing and midwifery programs at two universities in the eastern region of Poland were invited to participate in the study (Medical University of Lublin: 860; Medical University of Białystok: 680). The response rate was 37.1%, with 570 respondents. The age range of respondents was 19–57 years (median 25 years). There were 544 female respondents and 26 male respondents.

### Data collection procedure

The data was collected through a paper survey using the PAPI (Paper and Pencil Interviewing) method in the period between April and June 2023. The interviewers explained the aim of the study to participants and gave them instructions on how to complete the survey. Each study participant was provided with a survey form and an informed consent form. While filling out the survey, respondents could ask the interviewer questions. Participants completed the research by placing filled-in surveys in ballot boxes, which were opened after the interviewer left the room. Respondents' participation in the research was entirely voluntary and anonymous.

After having reviewed the questionnaires, 570 of the 586 completed surveys were qualified.

### Measures

**Socio-demographic variables.**   The personal data form consisted of socio-demographic questions such as age, gender (Male, Female, Other), year of college (first, second, third year of bachelor's studies and first and second year of studies, master's studies).

Other questions included:

- Sexual orientation: Heterosexual; Bisexual; Homosexual; Other

- Religious denomination: Catholic; Orthodox; Protestant; Islam; Judaism; Other

**The students' attitudes towards addressing sexual health extended questionnaire (SA-SH-Ext).**   The SA-SH-Ext questionnaire, which is based on the original SA-SH questionnaire [23, 24], is designed to measure students' attitudes towards sexuality-related matters in

their future profession and is intended for students of health and social care professions, such as occupational therapy, prosthetics and orthotics, nursing, physiotherapy and social work.

The SA-SH-Ext questionnaire consists of 27 items, covering the same 4 domains as the original SA-SH: "Feelings of comfortableness" (questions 1–13), "Fear of negative influence on future client relations" (questions 14–19), "Future working environment" (questions 20–22), and "Education needs" (questions 23–27) [24]. All items are measured by using a 5-step Likert scale (disagree, partly disagree, partly agree, agree, strongly agree). The responses "strongly agree and/or agree" are considered positive for positively loaded items, and for negatively loaded items, the responses "disagree and/or partly disagree" are considered as showing a negative attitude [23]. The response "partly agree" is not taken into account as it may be positive or negative, as the SA-SH Ext response categories clearly show response discrimination in the Rasch analysis [25]. Items 13–18 and 20–22 are reversed for analysis since these items are phrased negatively compared to all other items in the original SA-SH Ext questionnaire.

## Translation methods and language and cross-culture adaptation

The SA-SH Ext questionnaire's linguistic and cultural adaptation process was implemented in several stages. The Polish linguistic and cultural version of the SA-SH Ext questionnaire was created with the permission of Kristina Areskoug Josefsson, the copyright owner (permission obtained by email on 10th January 2023).

The World Health Organization's [26] protocol for scale translation was used:

1. Forward translation: two translators (Polish and English bilinguals), whose native language was Polish, independently translated the SA-SH Ext questionnaire from English to Polish in order to ensure conceptual equivalence of the scale items.

2. Expert panel: a bilingual panel of expert translators collaborated to create one common, compatible version of the SA-SH Ext questionnaire in Polish by sharing their independent translations. A consensus was reached to address differences between the two versions, and notes were made to record the concerns expressed and how they were handled.

3. Back-translation: The SA-SH Ext questionnaire in Polish was sent to two separate translators who had never seen the original SA-SH Ext. They then independently translated the Polish version back into English. These were English-speaking individuals who were also fluent in Polish.

4. Pretesting: This was carried out in a pilot study involving 30 target group nursing and midwifery students. After having completed the SA-SH Ext questionnaires, students were asked to participate in a discussion to identify any terms or expressions that they did not fully understand. Minor comments reported by students during the discussion were recorded on site. The translated version of the SA-SH Ext questionnaire in Polish was further refined through additional discussions with the translators, using notes and the group of students' observations from the pilot study.

5. Validation of psychometric properties. The validation of psychometric properties followed the stage of language and cultural adaptation of the questionnaire. The psychometric properties of SA-SH Ext were evaluated after 570 individuals completed the questionnaire.

**Ethical requirements.**    The research protocol was accepted by the Bioethics Commission of the Medical University
of Bialystok (ethical approval number: APK.002.181.2023). All the students participating in

the study were fully informed about the study, and their informed verbal consent was given. At the beginning of the survey, study participants received the following information: "The survey is anonymous, and participation in the study is voluntary. The information collected will only be used for collective scientific research. It is possible to resign from the study at any stage. Completing the survey is tantamount to agreeing to participate in the study." The decision to obtain verbal consent for participation in the study was undertaken in this way to protect respondents' anonymity and privacy avoiding additional documents requiring their signature. During the data collection, a researcher was available if any doubts or questions emerged.

**Statistical analysis.**    The reliability assessment was performed by measuring internal consistency reliability with Cronbach's alpha, with a Cronbach's alpha of 0.70−0.95 considered as good [27]. Construct validity was assessed via explorative factor analysis (EFA), with principal component as the extraction method. For factor rotations, Varimax with Kaiser normalization was applied. Items with high factor loadings defined each dimension. To be a clinically meaningful item in one of these factors, each item had to have a loading over 0.50 [28].

Each item was referred to the factor in which it had the highest loading. A scree plot was used to determine the optimal number of factors. Both the Kaiser-Meyer-Olkin measure of sampling adequacy and Bartlett test of sphericity were employed so as to ensure the usefulness of factor analysis for the collected data. A value close to 1.0 on the Kaiser-Meyer- Olkin measure of sampling adequacy indicated the proportion of variance of variables that might be caused by underlying factors, and a value below 0.05 on the Bartlett test of sphericity was used as a level to indicate that factor analysis was suitable.

The gender and field of study differences were measured by means of the non-parametric Mann Whitney U test. The year of study, sexual orientation, and religious denomination were assessed using the Kruskal-Wallis ANOVA test, while age was measured via Spearman's rank correlation.

The limit of statistical significance was set at $\alpha = 0.05$. The statistical analyses were performed with Statistica 13.3.

# Results

## Characteristics of participants

Table 1 presents the characteristics of the study group. The study involved 570 students, of which 72% were nursing students (n = 409) and 28% were midwifery students (28%, n = 161). The mean age in the study group was $\bar{x} = 24.3$, SD = 8.5 years. The majority of respondents were women (95%, n = 544), of heterosexual orientation (92%, n = 525), and Catholic denomination (83%, n = 469).

## Exploratory factor analysis—SA-SH Ext polish version (SA-SH Ext v.PL)

Internal consistency reliability with Cronbach's alpha showed very good results, with a Cronbach's alpha of 0.91 for the entire SA-SH-Ext v.PL scale. Cronbach´s alpha was between 0.60 and 0.95 even if any of the items were deleted. The internal consistency reliability test shows that all new items should be kept in the SA-SH-Ext v.PL questionnaire.

Construct validity was measured through explorative factor analysis and showed good results. The Kaiser-Meyer-Olkin measure of sampling adequacy (KMO) was high (0.923) and Bartlett test of sphericity was significant p<0.001. This means that the analysis presents a high explanatory level for variance in the responses. Fig 1 presents the scree plot, and reveals that the factor analysis led to 5 factors.

**Table 1. Characteristics of the study group.**

| Variables | Categories | N (%) |
|---|---|---|
| **Age [year]**<br>**M, SD, Min, Max** | Mean (M) | 24,3 |
| | Standard deviation (SD) | 8,5 |
| | Min—Max | 19–57 |
| **Gender** | Woman | 544 (95.0) |
| | Man | 26 (5.0) |
| **Profession** | Nursing students | 409 (72.0) |
| | Midwifery students | 161 (28.0) |
| **Year of study** | 1st (Bachelor) | 134 (23.0) |
| | 2nd (Bachelor) | 186 (33.0) |
| | 3rd (Bachelor) | 125 (22.0) |
| | 1st (Masters) (fourth year of studies) | 55 (10.0) |
| | 2nd (Masters) (fifth year of studies) | 69 (12.0) |
| **Religious** | Catholic | 469 (83.0) |
| | Orthodox | 31 (5.0) |
| | Other | 70 (12.0) |
| **Sexual orientation** | Heterosexual | 525 (92.0) |
| | Bisexual | 21 (4.0) |
| | Homosexual | 13 (2.0) |
| | Other | 11 (2.0) |

Based on the scree plot, 5 factors were distinguished. Table 2 shows the loading of the items in the factors. Items 1–12 were in factor 1, items 13–17 in factor 2, items 20–22 in factor 3, items 24 and 27 in factor 4, items 23 and 25, 26 in factor 5. Items 18 and 19 did not load in any

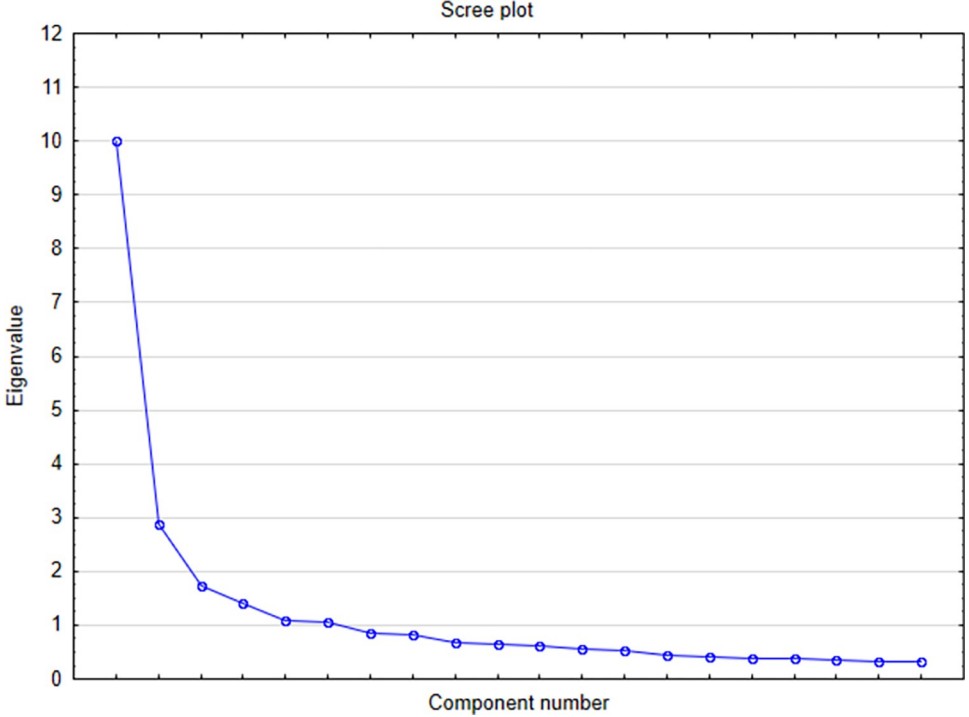

**Fig 1. The scree plot of factors SA-SH Ext v.PL.**

**Table 2. Factor analysis results.**

| No. | Items | Factor 1 | Factor 2 | Factor 3 | Factor 4 | Factor 5 |
|---|---|---|---|---|---|---|
| 1. | I feel comfortable about informing future clients about sexual health. | **0,7230** | 0,1445 | 0,0555 | 0,2714 | 0,1425 |
| 2. | I feel comfortable about initiating a conversation regarding sexual health with future | **0,7820** | 0,1753 | 0,0504 | 0,1371 | 0,1733 |
| 3. | I feel comfortable about discussing sexual health with future clients. | **0,8058** | 0,1829 | 0,0814 | 0,2443 | 0,1299 |
| 4. | I feel comfortable about discussing sexual health issues with future clients with physical disability. | **0,7814** | 0,1534 | 0,0806 | 0,1455 | 0,1120 |
| 5. | I feel comfortable about discussing sexual health issues with future clients with physical disease. | **0,8509** | -0,0048 | 0,1561 | -0,0536 | 0,0558 |
| 6. | I feel comfortable about discussing sexual health issues with future clients with intellectual disability. | **0,8626** | 0,0266 | 0,1652 | -0,0689 | 0,0804 |
| 7. | I feel comfortable about discussing sexual health issues with future clients with mental illness. | **0,8264** | 0,0107 | 0,1294 | -0,1334 | 0,0905 |
| 8. | I feel comfortable about discussing sexual health issues with future clients, regardless of their sex. | **0,7645** | 0,0119 | 0,1315 | -0,0830 | 0,1027 |
| 9. | I feel comfortable about discussing sexual health issues with future clients, regardless of their age. | **0,7625** | 0,1213 | 0,0511 | 0,0641 | 0,1470 |
| 10. | I feel comfortable about discussing sexual health issues with future clients, regardless of their cultural background. | **0,7673** | 0,0976 | 0,1211 | 0,1213 | 0,1084 |
| 11. | I feel comfortable about discussing sexual health issues with future clients, regardless of their sexual orientation. | **0,7448** | 0,0700 | 0,1480 | 0,1984 | 0,0612 |
| 12. | I feel comfortable about discussing specific sexual activities with future clients. | **0,7558** | 0,2265 | 0,0608 | 0,0861 | 0,2158 |
| 13. | I am unprepared to talk about sexual health with future clients. | 0,3346 | **0,5023** | 0,0136 | 0,1528 | 0,3617 |
| 14. | I believe that I might feel embarrassed if future clients talk about sexual issues. | 0,3953 | **0,6120** | 0,0826 | 0,2356 | 0,1276 |
| 15. | I believe that future clients might feel embarrassed if I bring up sexual issues. | -0,0535 | **0,6884** | 0,2030 | -0,3843 | 0,0567 |
| 16. | I am afraid that future clients might feel uneasy if I talk about sexual issues. | 0,0504 | **0,7202** | 0,1654 | -0,3557 | 0,0622 |
| 17. | I am afraid that conversations regarding sexual health might create a distance between me and the clients. | 0,1650 | **0,6423** | 0,2288 | 0,0697 | -0,0769 |
| 18. | I believe that I will have too much to do in my future profession to have time to handle sexual issues. | 0,1408 | -0,0899 | 0,4372 | 0,0776 | -0,0673 |
| 19. | I will take time to deal with clients' sexual issues in my future profession. | 0,3826 | -0,0224 | 0,3553 | 0,2674 | 0,4084 |
| 20. | I am afraid that my future colleagues would feel uneasy if I brought up sexual issues with clients | 0,0993 | 0,3255 | **0,7428** | 0,0339 | 0,0481 |
| 21. | I am afraid that my future colleagues would feel uncomfortable dealing with questions regarding clients' sexual health. | 0,1003 | 0,3115 | **0,7707** | -0,0632 | 0,1034 |
| 22. | I believe that my future colleagues will be reluctant to talk about sexual issues. | 0,1624 | 0,2599 | **0,6303** | 0,0521 | 0,2354 |
| 23. | In my education I have been educated about sexual health. | 0,0709 | -0,0063 | 0,1336 | -0,1567 | **0,7619** |
| 24. | I think that I, as a student, need to get basic knowledge about sexual health in my education | 0,2082 | 0,0087 | 0,0900 | **0,7387** | 0,0929 |
| 25. | I have sufficient competence to talk about sexual health with my future clients. | 0,3828 | 0,1684 | 0,1584 | 0,4279 | **0,5413** |
| 26. | I believe in my own ability to promote sexual health in my future profession. | 0,3362 | 0,1107 | -0,0532 | -0,0723 | **0,7520** |
| 27. | I think that I need to be trained in my education to talk about sexual health. | -0,0278 | -0,2165 | 0,0480 | **0,7033** | -0,2743 |
| **Alpha–Cronbach Coefficient** | | 0,9 | 0,7 | 0,8 | 0,6 | 0,7 |
| **Variance** | | 8,25 | 2,59 | 2,19 | 1,97 | 2,11 |
| **% of variance** | | 0,31 | 0,10 | 0,08 | 0,07 | 0,08 |

of the factors. The factors explain 64% of the variances in the factor analysis, which is a reasonable result.

## Students' attitudes toward addressing patient sexual health and the relationship between SA-SH Ext v.PL and professional and socio-demographic variables

Table 3 presents descriptive statistics for *SA-SH Ext v.PL*. The mean value for SA-SH Ext v.PL was M = 3.19 SD = 0.39. The highest mean value was demonstrated by "Educational needs—Awareness of knowledge gap" item (M = 4.12, SD = 0.66), and the lowest was indicated by "Educational needs—Awareness of the need for competences" (M = 2.92, SD = 0.74). The modal value indicates the most frequently repeated result. Herein, the most common result

**Table 3. Descriptive statistics SA-SH Ext v.PL.**

| SA-SH Ext v.PL subscales | M | Mo | N$_{Mo}$ | SD | V. | SKE |
|---|---|---|---|---|---|---|
| Present feelings of comfortableness | 3.15 | 3 | 44 | 0.80 | 25.52 | 0.18 |
| Future working environment—awareness | 3.23 | 3 | 77 | 0.66 | 20.73 | 0.02 |
| Fear of negative influence on future patient relation | 3.02 | 3 | 124 | 0.82 | 27.26 | -0.09 |
| Educational needs—Awareness of knowledge gap | 4.12 | 4 | 195 | 0.66 | 16.13 | -0.52 |
| Educational needs—Awareness of the need for competences | 2.92 | 3 | 120 | 0.74 | 25.48 | 0.00 |
| Total | 3.19 | 3 | 20 | 0.39 | 12.21 | 0.32 |

Note. Students' attitudes towards sexual health are measured on a 5-point Likert scale ranging from 5 = *strongly agree* to 1 = *disagree*.

M—mean value, Mo–modal value, SD- standard deviation; V- coefficient of variation; SKE–skewness coefficient

was 3 points in the scales, except for the notion "Educational needs—Awareness of knowledge gap", where the modal value was 4. Over all, there were as many as 195 such values. The "Fear of negative influence on future patient relations" item showed the widest range of results–M = 3.02 SD = 0.82, and the coefficient of variation was 27.26%. In the case of the first two and last two subscales, the asymmetry is right-sided, i.e., most of the results are below the mean value, while in the "Fear of negative influence on future patient relations" and "Educational needs—Awareness of knowledge gap" scales, the asymmetry is left-sided, which means that most of the results had a value higher than mean value (M = 3.19).

Additionally, all significant variations in the assessment of each subscale with respect to gender, field of study, year of study, sexual orientation, and religious denomination were examined. Table 4 shows the results of the difference analysis.

The obtained results show that none of the five subscales' assessments are affected by sexual orientation. The year and field of study, however, significantly differentiate the assessments in all subscales. Gender differentiates the "Present feelings of comfortableness" (p = 0.0026) and "Fear of negative influence on the future patient relations" item (p = 0.0155)—Table 4. Men feel more comfortable (3.63), but they also have greater concerns about future relationships (3.44)–Fig 2.

The field of study has an impact on the assessment of all subscales, that is "Present feelings of comfortableness" (p = 0,0000), "Future working environment" (p = 0,0000), "Fear of

**Table 4. Results of the analysis of the SA-SH Ext v.PL subscales and professional, socio-demographic variables.**

| SA-SH Ext subscales | Gender | Profession | Year of study | Sexual orientation | Religious |
|---|---|---|---|---|---|
| | | | p-value | | |
| Present feelings of comfortableness (items 1–12) | 0.0026** | 0.0000*** | 0.0000*** | 0.3656 | 0.0018** |
| Future working environment (items 13–17) | 0.8713 | 0.0000*** | 0.0001** | 0.1248 | 0.1480 |
| Fear of negative influence on future patient relations (items 20–22) | 0.0155* | 0.0000*** | 0.0000*** | 0.5619 | 0.0622 |
| Educational needs: Awareness of knowledge gap (items 24 and 27) | 0.1091 | 0.0000*** | 0.0000*** | 0.4949 | 0.0261* |
| Educational needs: Awareness of the need for competences (items 23, 25, 26) | 0.4801 | 0.0000*** | 0.0003* | 0.9011 | 0.1132 |

*p< 0.05

**p< 0.01

***p< 0.001

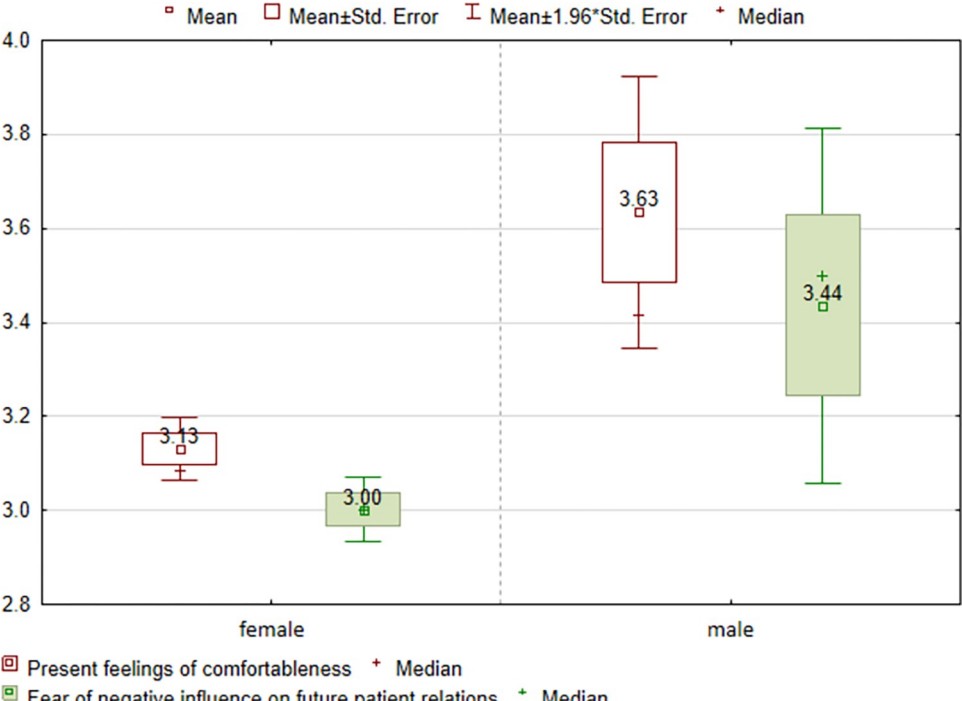

**Fig 2. Assessment of the "Present feelings of comfortableness" and "Fear of negative influence on future patient relations" item broken down by gender.**

negative influence on future patient relations" (p = 00000), "Educational needs—Awareness of knowledge gap" (p = 0,0000) and "Educational needs - Awareness of the need for competences" (p = 0,0000)—Table 4. Nurses have greater concerns associated with the "Future working environment" (3.83) and have greater "Fear of negative influence on future patient relations" (3.17), while midwives chose "Present feelings of comfortableness" higher (3.50) and see greater "Educational needs" (4.40)–Fig 3.

Year of study differentiated the scores of all five subscales—Table 4. First-year respondents gave the highest rating (3.56) for the "Present feelings of comfortableness" item, while fifth-year respondents gave this the lowest rating (2.82). Fourth-year students (3.36) had the most concerns about the "Future working environment" item, while first-year students (3.00) had the least concerns. Fourth-year respondents reported the highest rate (3.25) on the "Fear of negative influence on future patient relations" item, while first-year respondents reported the lowest rate (2.67). First-year respondents perceived greater "Educational needs" most frequently (3.17).—Fig 4.

Religious denomination was correlated with the "Present feelings of comfortableness" (p = 0,0018) and the "Educational needs—Awareness of knowledge gap" item (p = 0,0261). Those in the "other" group are most comfortable (4.24), while "orthodox" (3.89) are least comfortable. There is a similar assessment of the educational needs—Fig 5.

Age was related to scores with regard to all subscales. The results indicate that the "Present feelings of comfortableness" item is most influenced by age, with feeling of comfort declining as age increases (R = -0.28, p< 0.01). Similarly, in the case of the "Educational Needs" item, the coefficient is also negative, but slightly smaller (R = -0.21 p< 0.01 and R = -0.12 p< 0.01). The scores for the "Fear of a negative impact on future patient relations" and "Future working

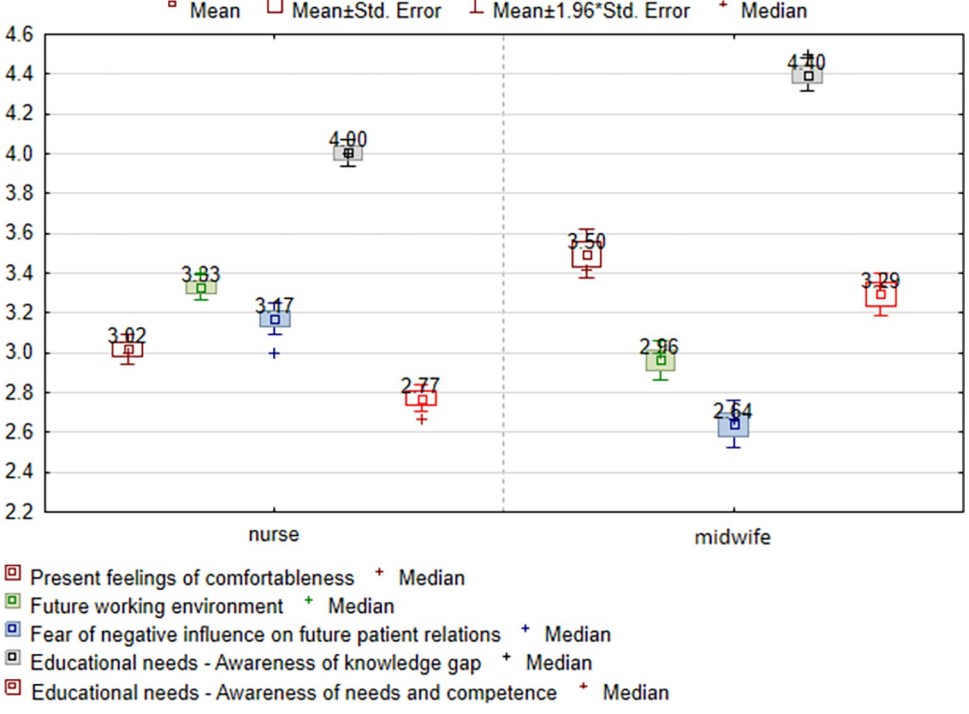

**Fig 3. Assessment of the SA-SH Ext subscales in relation to field of study.**

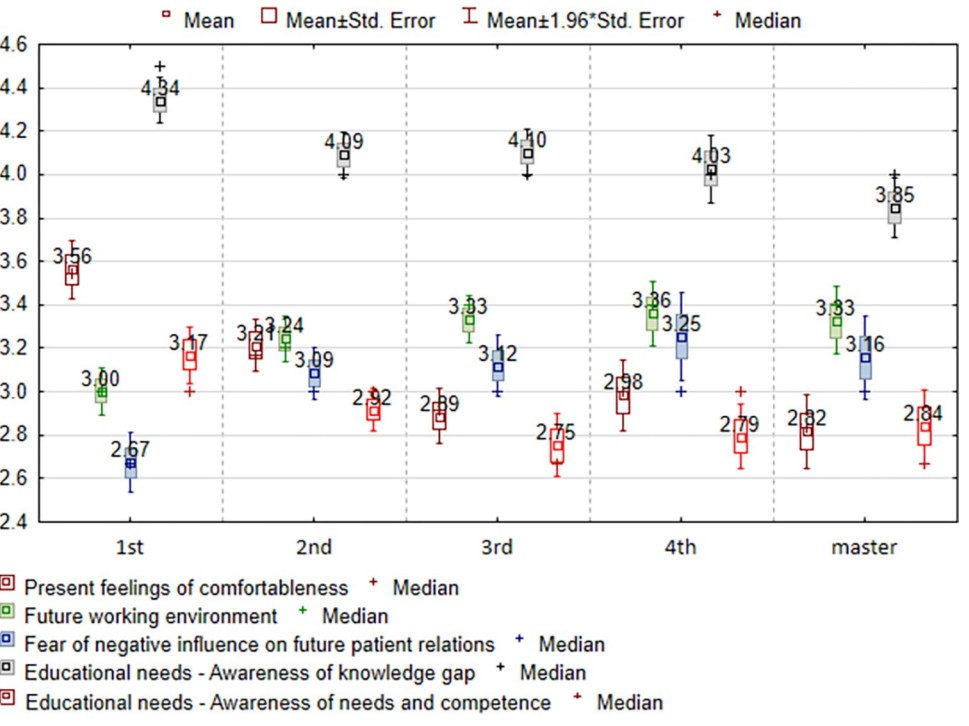

**Fig 4. Assessment of the SA-SH Ext subscales in relation to the year of study.**

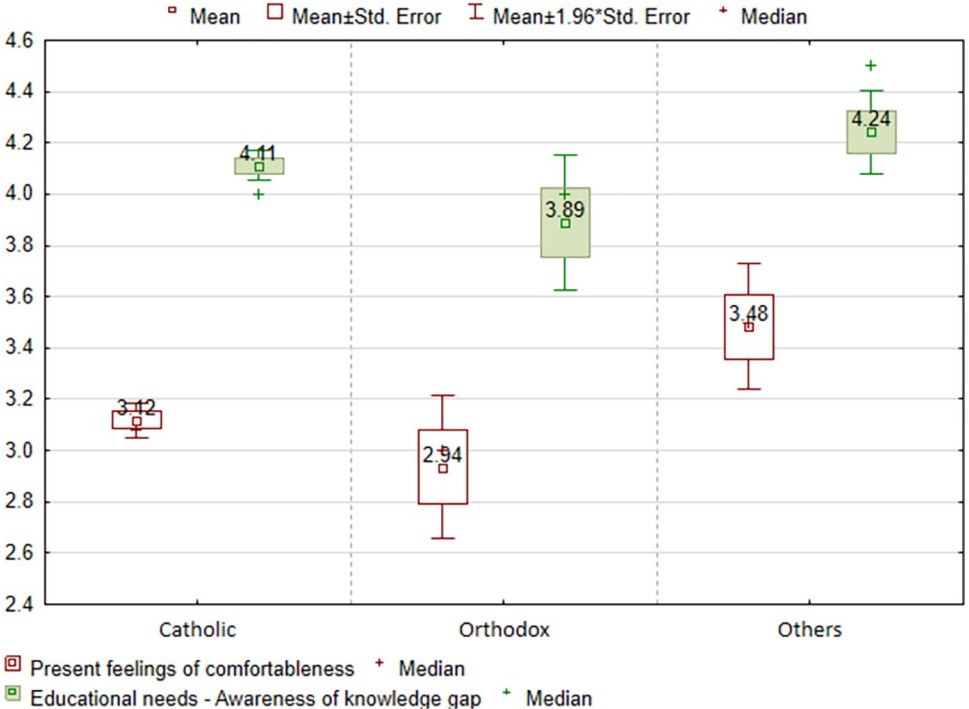

**Fig 5. Assessment of the SA-SH Ext subscales in relation to religious denomination.**

environment" items increase with age, indicating a positive correlation between the two subscales (Table 5).

## Discussion

### Assessment of psychometrics of the SA-SH-Ext v.PL questionnaire

The main aim of this study was to assess the effectiveness of a translation of the SA-SH Ext questionnaire from English into Polish through making necessary cultural/linguistic adaptations, and to determine the validity and reliability of the Polish version of the questionnaire. We were the first to obtain a Polish questionnaire that assessed the attitudes of nursing and midwifery students regarding patient sexual health. The results of this study, which measures the validity and reliability of the SA-SH-Ext v.PL questionnaire show that this instrument has

**Table 5. Spearman's rank correlation coefficients between age and the assessment of the SA-SH Ext v.PL subscales.**

| Variables | R | t(N-2) | p |
|---|---|---|---|
| Age & Present feelings of comfortableness | -0.28 | -6.88 | 0.0000*** |
| Age & Future working environment | 0.12 | 2.81 | 0.0051* |
| Age & Fear of negative influence on future patient relations | 0.21 | 5.00 | 0.0000*** |
| Age & Educational needs—Awareness of knowledge gap | -0.21 | -5.18 | 0.0000*** |
| Age & Educational needs—Awareness of the need for competences | -0.12 | -2.85 | 0.0045** |

*p< 0.05

**p< 0.01

***p< 0.001; R–Spearman's rank correlation coefficient

appropriate psychometric properties and is helpful in measuring students' attitudes towards patient sexual health care.

Cronbach's total alpha value for the SA-SH Ext v.PL was 0.91, while the dimensions in each of the 5 subscales were as follows: $\alpha = 0.9$, $\alpha = 0.7$; $\alpha = 0.8$; $\alpha = 0.6$ and $\alpha = 0.7$. In the literature, Cronbach's alpha values of α = 0.61–0.80 indicate moderate reliability, while values in the range of α = 0.81–0.95 indicate high reliability [27]. For the original SA-SH questionnaire of Areskoug-Josefsson, Juuso, Gard et al. [23] the Cronbach's alpha value is $\alpha = 0.67$, and in the study conducted by Areskoug-Josefsson, Thidell, Rolander et al. [29] the Cronbach's alpha value is $\alpha = 0.86$. In accordance with the literature, a comparison of the Polish version of the SA-SH Ext questionnaire with the original scale demonstrates that it is highly reliable and comparable to the original scale [30]. Additionally, other countries and care environments have confirmed the SA-SH questionnaire's usefulness [23, 25, 31–36], which also determines the potential value of this instrument.

An exploratory factor analysis shows that the domains for SA-SH-Ext v.PL are not completely consistent with the factor analysis of the original SA-SH, especially with respect to factor 4 [23]. The analysis of construct validity in the Polish version of the questionnaire distinguished five major factors: "Present feelings of comfortableness"; "Future working environment", "Fear of negative influence on future patient relations", "Educational needs—Awareness of knowledge gap" and "Educational needs—Awareness of the need for competence". Factor 1 "Present feelings of comfortableness" includes items 1–12; Factor 2: "Future working environment" includes items 13–17, Factor 3: "Fear of negative influence on future patient relations" includes items 20–22. However, Factor 4 was extracted from the original Factor 4 "Educational needs" in the polarisation of the results: "Educational needs—Awareness of knowledge gap and Educational needs" (items 23 and 25) and Factor 5: "Educational needs—Awareness of the needs for competence" (items 23 and 25, 26). Item 18 (*I believe that I will have too much to do in my future profession to have time to handle sexual issues*) and item 19 (*I will take time to deal with clients' sexual issues in my future profession*) did not load in any of the factors. One of these items (item 19) was also not loaded into the main factors in the original SA-SH questionnaire [23]. However, it was decided that it should be left in the SA-SH based on prior research and practical experience. In accordance with the authors' prior experience with SA-SH, items 18 and 19 were incorporated into the factors where they most fit. Additionally, items 18 and 19 were kept as they are in line with earlier studies that described competence and time as causes for failure to have discussions with patients about sexual health and the recognition of a need to be competent in the future profession [4, 5]. However, in order to ensure the stability of the factors, a factor analysis is advised for future studies and practical application in different settings.

Taking into consideration previous studies of the original SA-SH questionnaire [23, 29, 37] and the Danish version of SA-SH [3, 38], the SA-SH Ext v.PL questionnaire is characterised by high reliability and validity. The SA-SH-Ext v.PL questionnaire may be a helpful instrument for assessing educational interventions related to patient sexual health in Polish nurse and midwife education curriculum.

## Nursing and midwifery students' attitudes towards addressing patient sexual health according to the SA-SH-Ext v.PL questionnaire

The aim of this study was to determine the attitudes of nursing and midwifery students towards patient sexual health in their future profession. This is the first research to provide basic information on the knowledge, attitudes, and preparedness of nursing and midwifery students in Poland in the care of patient sexual health.

According to the study results, students' attitudes towards patient sexual health in their future profession were deemed to be moderately positive. This is primarily indicated by the highest average subscale scores: Factor 2: "Future working environment" (M = 3.23; SD = 0.66) and Factor 1: "Present feelings of comfortableness" (M = 3.15; SD = 0.80), also expressed by a high awareness of Factor 4: "Educational needs—Awareness of knowledge gap" (M = 4.12, SD = 0.66) and the lowest score for Factor 5: "Educational needs—Awareness of the need for competences" (M = 2.92, SD = 0.74). The results in our study obtained lower values in comparison with the sub-scale averages in a group of nursing students in the USA (n = 159) in a study undertaken by Russell, Chen., Jensen et. al. [39], herein: Factor 1 is M = 3.68, SD = 0.83, Factor 2 is M = 4.15, SD = 0.69, Factor 3 is M = 3.54, SD = 0.70, and Factor 4 is M = 3.92, SD = 0.49.

A review of the currently available studies aimed at determining nursing and midwifery students' attitudes towards addressing patients' sexual health according to the SA-SH questionnaire reveals a diversity of results, ranging from positive attitudes [3, 40, 41], to the prevalence of negative attitudes among students [36, 42]. In addition, all subscales of the SA-SH Ext v.PL questionnaire demonstrate significant differences in the preparedness of nursing and midwifery students (p = 0.0000). Nursing students rated the "Future working environment" (M = 3.23) and "Fear of negative influence on future patient relations" (M = 3.02) higher, while midwifery students rated "Present feelings of comfortableness" higher (M = 3.43) and see greater "Educational needs" (M = 3.74) to be of relevance.

In a study of midwifery students' attitudes, beliefs, and comfort level regarding sexual counselling in Turkey (n = 650, using the Sexuality Attitudes and Beliefs Survey—SABS), it was found that, despite midwifery students' positive attitudes toward patients' sexual relationships and their importance in marital life, they lacked adequate skills in providing sexual health counselling to women [43].

A systematic review of nursing students' competence in sexual health care by Blakey & Aveyard [44] identified several important issues, among others, nursing students' positive attitudes toward patient sexual health care. However, many students stated that they felt uncomfortable to start a conversation or bring up sexual health-related topics. In clinical and professional education, many nursing students also lacked mentor role models and knowledge about sexual health [44].

Numerous authors have long examined this crucial subject in their empirical analyses of nursing students' knowledge, attitudes, and preparedness to work with patients who have sexual health issues [17, 19, 45]. For the comfort of patients and the delivery of high-quality health care, nurses' and midwives' attitudes toward patient sexuality are crucial. According to the results of a multi-centre study on attitudes and beliefs regarding the management of sexual health in primary health care, physicians and nurses believed that patients were uncomfortable discussing sex-related matters with them. Additionally, significant differences were observed between the professions, with nurses being more likely to be asked by patients about sexual health and to claim to have the necessary knowledge to appropriately respond to patients' inquiries [46].

The existence of a valuable and valid questionnaire in assessing future nursing and midwifery professionals' attitudes towards patient sexual health issues will be helpful in order to better address specific patient needs or knowledge gaps, such as sexual health issues of, for instance, people with disabilities or people having different sexual orientations.

## Strengths and weaknesses of the study

The SA-SH Ext v.PL questionnaire is a research instrument that is reliable and relevant in a local environment. Since this study was carried out in multiple academic centres, its results

can be broadly applied, thus demonstrating the strength of this research. A limitation of the current study is its exclusive focus on nursing and midwifery students. It is worthwhile to conduct further research taking into account students from different fields of study, such as medicine, physiotherapy, dietetics, and so on, in order to create a multidisciplinary medical team. The subsequent limitation of the study may be the use of an individual reporting instrument, which raises the possibility that respondents may not have answered honestly to sensitive questions that dealt with issues considered to be of a delicate nation.

## Implications for practice

Professional education needs to change to incorporate modules on patient sexual health into the curriculum and to encourage awareness of equal gender roles in clinical practice in order to improve the attitudes of nursing and midwifery students. Holistic sexual health education should be provided to professional nurses and midwives. In order to support coeducation and the development of healthy sexual behaviour (including gender equality, sex, social equality, erotophilic behaviour and sociosexuality), as well as the prevention of sexually transmitted diseases, significant somatic and mental health issues related to sexual activity and risky sexual behaviour, sex education curriculum in higher education should cover the complexity of issues related to the development of human sexuality and care, starting with school medicine.

## Conclusions

Based on the study of validity and reliability of the SA-SH Ext v.PL questionnaire, it was determined that the scale is appropriate for nursing and midwifery students. In addition, it was found that the questionnaire is a valid and reliable measurement instrument in assessing students' attitudes towards patient sexual health. The analysis of construct validity demonstrated five major factors: "Present feelings of comfortableness", "Future working environment", "Fear of negative influence on future patient relations", "Educational needs—Awareness of knowledge gap", "Educational needs—Awareness of the needs for competences". The construct validity and internal consistency reliability of the SA-SH Ext v.PL scale showed very good results.

In addition, it should be noted that nursing and midwifery students had moderately positive attitudes towards patient sexual health in their future profession. The "Educational needs—Awareness of knowledge gap" item was rated higher, while the "Awareness of the need for competences" item was rated lower. Incorporating sexual health education and research projects into undergraduate vocational education seems to be advantageous for students' knowledge and research findings, as well as for fostering deeper reflection on their attitudes, knowledge, and needs related to sexual health competence in their future careers.

## Supporting information

**S1 Dataset.**
(XLS)

## Acknowledgments

We acknowledge support from the *Faculty of Health Sciences*, *Medical University of Białystok and Faculty of Health Sciences*, *Medical University of Lublin* and our dear participants who gave us their honest views about the nursing and midwifery students' attitudes towards addressing patient sexual health in their future profession.

## Author Contributions

**Conceptualization:** Barbara Ślusarska, Ludmiła Marcinowicz.

**Data curation:** Barbara Ślusarska, Ludmiła Marcinowicz.

**Formal analysis:** Barbara Ślusarska, Ludmiła Marcinowicz.

**Funding acquisition:** Barbara Ślusarska, Ludmiła Marcinowicz.

**Investigation:** Barbara Ślusarska, Ludmiła Marcinowicz.

**Methodology:** Barbara Ślusarska, Ludmiła Marcinowicz.

**Project administration:** Barbara Ślusarska, Ludmiła Marcinowicz.

**Supervision:** Ludmiła Marcinowicz.

**Validation:** Barbara Ślusarska.

**Writing – original draft:** Barbara Ślusarska.

**Writing – review & editing:** Ludmiła Marcinowicz.

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
