## [Decision Letter · Decision Letter 0]

21 Dec 2023

PONE-D-23-36317Nursing and Midwifery Students’ Attitudes Towards Addressing Patient Sexual Health in their Future Profession: Polish Adaptation and Validation of the Students’ Attitudes Towards Addressing Sexual Health Extended Questionnaire (SA-SH-Ext)PLOS ONE

Dear Dr. Marcinowicz,

Thank you for submitting your manuscript to PLOS ONE. After careful consideration, we feel that it has merit but does not fully meet PLOS ONE’s publication criteria as it currently stands. Therefore, we invite you to submit a revised version of the manuscript that addresses the points raised during the review process.

We look forward to receiving your revised manuscript.

Kind regards,

Nabeel Al-Yateem, PhD

Academic Editor

PLOS ONE

Journal Requirements:

Reviewers' comments:

Reviewer's Responses to Questions

**Comments to the Author**

1. Is the manuscript technically sound, and do the data support the conclusions?

Reviewer #1: Yes

Reviewer #2: Yes

2. Has the statistical analysis been performed appropriately and rigorously? 

Reviewer #1: I Don't Know

Reviewer #2: Yes

3. Have the authors made all data underlying the findings in their manuscript fully available?

Reviewer #1: Yes

Reviewer #2: Yes

4. Is the manuscript presented in an intelligible fashion and written in standard English?

Reviewer #1: Yes

Reviewer #2: Yes

5. Review Comments to the Author

Reviewer #1: Dear Editor

Thank you for your invitation to review of manuscript entitled " Nursing and Midwifery Students’ Attitudes Towards Addressing Patient Sexual Health

in their Future Profession: Polish Adaptation and Validation of the Students’ Attitudes

Towards Addressing Sexual Health Extended Questionnaire (SA-SH-Ext)

This is a strong introduction that effectively sets the stage for your research paper on sexual health care education for nurses and midwives in Poland. It highlights the importance of sexual health, the challenges faced by healthcare professionals in providing this care, and the need for adequate education.

Introduction

comment 1: Reduce redundancy. The first paragraph could be shortened by removing some redundant information about the importance of sexual health.

comment 2: Provide more specific details about your research: While you mention the need to assess students' competence and preparedness, you could provide a more specific outline of your research methods and objectives.

comment 3: Consider adding a thesis statement: A clear thesis statement at the end of the introduction would help to summarize your main argument and provide a roadmap for the rest of the paper.

comment 4: You could consider mentioning any relevant government policies or regulations in Poland concerning sexual health education for healthcare professionals.

comment 5: You could provide a brief overview of the literature review you conducted, highlighting the key findings.

comment 6: You could briefly discuss the potential implications of your research for improving sexual health care in Poland.

Material and Method

comment 1: While you describe the sampling method used, it would be helpful to explain why convenience sampling was chosen rather than a more random approach.

comment 2: Missing information about data analysis: While you mention validating the psychometric properties of the SA-SH-Ext, you do not provide details about the specific statistical methods used for this analysis.

comment 3: Consider adding a table or figure: A table summarizing the characteristics of the participants and a figure depicting the structure of the SA-SH-Ext questionnaire would further enhance clarity.

Discussion

comment 1: You mention that two items (18 and 19) did not load significantly onto any of the factors. While you explain why they were kept in the questionnaire, you could consider conducting further research to explore their validity and reliability in the Polish context.

comment 2: You briefly mention the potential for the SA-SH-Ext v.PL questionnaire to be used with students from other disciplines. You could consider elaborating on this point and discussing how the questionnaire could be adapted for use in other healthcare professions.

comment 3: You acknowledge the potential for bias due to self-reporting. You could consider discussing alternative data collection methods, such as focus groups or interviews, that might be less susceptible to this bias.

Reviewer #2: First of all, congratulations for the work done, the research study and the tool to be validated are of great interest at present and the study is well justified. Congratulations for the results, the psychometric characteristics determined are powerful.

As a reviewer, a doubt arises in my mind, have you not considered completing the study with a confirmatory factor analysis (CFA)?

After reviewing the manuscript, the following points could be improved:

1. The introduction could be improved, I think it should be pointed out if the tool has been validated in other languages and provide data on these validations.

2. It is also possible to address cultural differences if any in the introduction or discussion.

3. On the other hand, you talk about validation in general without a gender perspective. Is this validation valid for male students? Because of the sample, and the scarcity of male students, I think it is only possible to talk about the tool being validated for female students.

4. It should be indicated, if considered, the need to perform the validation in male population, as a future study.

6. PLOS authors have the option to publish the peer review history of their article (what does this mean?). If published, this will include your full peer review and any attached files.

Reviewer #1: No

Reviewer #2: **Yes: **Ivan Santolalla-Arnedo

---

## [Author Response · Author response to Decision Letter 0]

4 Jan 2024

Dear Editor-in-Chief PLOS ONE

I express my appreciation to the reviewers and the editorial board for taking the time and effort to improve the work and provide insightful comments.

I am pleased to have been allowed to revise our manuscript ID: PONE-D-23-36317

entitled: Nursing and Midwifery Students’ Attitudes Towards Addressing Patient Sexual Health in their Future Profession: Polish Adaptation and Validation of the Students’ Attitudes Towards Addressing Sexual Health Extended Questionnaire (SA-SH-Ext)

Dear Madam or Sir Reviewer

Thank you for your valuable comments and efforts to improve our manuscript; the manuscript was revised based on your comments and recommendations.

All responses to Reviewers' comments are marked in blue in the text.

Reviewer comments

Response to Reviewer 1 Comments

Reviewer #1: Dear Editor

Thank you for your invitation to review of manuscript entitled " Nursing and MidwiferyStudents’ AttitudesTowards Addressing Patient Sexual Health in their Future Profession: Polish Adaptation and Validation of the Students’ Attitudes Towards Addressing Sexual Health Extended Questionnaire (SA-SH-Ext)

This is a strong introduction that effectively sets the stage for your research paper on sexual healthcare education for nurses and midwives in Poland. It highlights the importance of sexual health, the challenges faced by healthcare professionals in providing this care, and the need for adequate education. Response: Thank you for your positive opinion about the article.

Introduction

comment 1: Reduce redundancy. The first paragraph could be shortened by removing some redundant information about the importance of sexual health.

 Response: Thank you for your suggestion. The first paragraph has been revised and shortened

comment 2: Provide specific details about your research: While you mention the need to assess student's competence and preparedness, you could provide a more specific outline of your research methods and objectives

 Response: Thank you for your comment. A more detailed outline of the research methods and objectives is described in the last paragraph of the introduction.

comment 3: Consider adding a thesis statement: A clear thesis statement at the end of the introduction would help to summarize your main argument and provide a roadmap for the rest of the paper. Response: Thank you for your insight. We have tried to highlight the main argument to justify the action plan for the following parts of the article.

comment 4: You could consider mentioning any relevant government policies or regulations in Poland concerning education for healthcare professionals.

 Response:A description of Polish government regulations regarding the education of health care workers in the area of sexual health has been added

comment 5: You could provide a brief overview of the literature review you conducted, highlighting the key findings.

 Response: Thank you for your insight. We have added a short literature review

comment 6: You could briefly discuss the potential implications of your research for improving sexual healthcare in Poland.

 Response: Thank you for your attention. The potential implications of our research for improving sexual health care in Poland have been added in the Implications for practice section

Material and Method

comment 1: While you describe the sampling method used, it would be helpful to explain why convenience sampling was chosen rather than a more random approach.

 Thank you for your attention. Limited possibilities of financing research made it necessary to choose cheaper options based on a non-probabilistic method of selecting a sample of respondents with voluntary selection of respondents.

We expected it would be more challenging to keep respondent anonymity when using a random approach. We assumed that participation in the research was voluntary because of the delicate nature of the topic and the sensitive issues under study.

comment 2: Missing information about data analysis: While you mention validating the psychometric properties of the SA-SH-Ext, you do not provide details about the specific statistical methods used for this analysis.

Page 16 of the PDF version contained the statistical analysis. We have completed the description of “Statistical analysis” methods.

comment 3: Consider adding a table or figure: A table summarizing the characteristics of the participants and a figure depicting the structure of the SA-SH-Ext questionnaire would further enhance clarity.

. Figures and tables in the pdf version were on pages 32-39. They have now been included in the text of the article.

Discussion

comment 1: You mentioned (18 and 19) did not significantly of the factors. While you explain why they were kept in the questionnaire, you could consider conducting further research to explore their validity and reliability in the Polish context.

Response: Thank you for your insight. We have added and recognize the need to develop research on checking the validity and reliability of the SA SH Ext scale in the Polish context in the discussion part

comment 2: You briefly mention the potential for the SA-SH-Ext v.PL questionnaire to be used with students from other disciplines. You could consider elaborating on this point and discussing how the questionnaire could be adapted for use in other healthcare professions.

 Thank you for your attention. This correct observation was included in the discussion part of the article and in the “Strengths and weaknesses of the study” section

comment 3: You acknowledge the potential for bias due to self-reporting. You could consider discussing alternative data collection methods, such as focus groups or interviews, that might be less susceptible to this bias.

 Thank you for this important tip, we will consider using it in future research

Response to Reviewer 2 Comments

Reviewer #2: First of all, congratulations for the work done, the research study and the tool to be validated are of great interest, and the study is well justified. Congratulations on the results, the psychometric characteristics determined are powerful

 Response: Thank you for your positive opinion about the article.

As a reviewer, a doubt arises in my mind, have you not considered completing the study with a confirmatory factor analysis (CFA)?

 We employed exploratory factor analysis (EFA) because, given the data, it was necessary to identify the number of factors on an ad-hoc basis. When adapting the tool, we found this instrument particularly useful when working with a dataset containing multiple variables, the structure of which was not clearly defined.

Confirmatory factor analysis (CFA), which is frequently employed in structural equation modelling, makes assumptions about the number of factors to be included based on a-priori theoretical understanding, as well as on the variables that should be strongly associated with each factor. This type of factor analysis is frequently applied to validate new findings in the context of well-studied phenomenon structures. This is the case, for instance, when developing new assessment instruments to identify phenomena with a solid theoretical foundation.

Therefore, we are convinced that the CFA is not required in our case, especially since the SA-SH-Ext scale's parameters are satisfactory for the tool's psychometric evaluation when the EFA method is used.

After reviewing the manuscript, the following points could be improved:

1. The introduction could be improved, I think it should be pointed out if the tool has been validated in other languages and provide data on these validations.

Response: Thank you for your attention. We have added a short paragraph on the use of SA-SH Ext scale validation in other languages, and the validation results are discussed in the discussion

2. It is also possible to address cultural differences if any in the introduction or discussion.

 Response: Thank you for your insight. In the introduction, we drew attention to cultural differences, especially in relation to barriers and differences in approach to education and the health care system

3. On the other hand, you talk about validation in general without a gender perspective. Is this validation valid for male students? Because of the sample, and the scarcity of male students, I think it is possible to talk about the tool being validated for female students.

 Response: Thank you for your important observation.

Indeed, the study group consisted of only 26 men and 544 women. This may be considered a limitation of the study (indicated in the “Strengths and weaknesses of the study” section). However, the results show that gender differentiated “Present feelings of comfortableness” (p=0.0026) and “Fear of negative influence on future“ (p=0.0155) scores - Table 4. Men felt more comfortable (3.63), but they also had greater concerns about future relationships (3.44) -- Figure 2.

4. It should be indicated, if considered, the need to perform the validation in the male population, as a future study.

 Thank you for this important tip. We will consider using it in future research

Dear Reviewer, best efforts have been made to improve the manuscript, and I hope I have met your expectations. 

Thank you

---

## [Decision Letter · Decision Letter 1]

29 Feb 2024

Nursing and Midwifery Students’ Attitudes Towards Addressing Patient Sexual Health in their Future Profession: Polish Adaptation and Validation of the Students’ Attitudes Towards Addressing Sexual Health Extended Questionnaire (SA-SH-Ext)

PONE-D-23-36317R1

Dear Dr. Marcinowicz,

We’re pleased to inform you that your manuscript has been judged scientifically suitable for publication and will be formally accepted for publication once it meets all outstanding technical requirements.

Kind regards,

Nabeel Al-Yateem, PhD

Academic Editor

PLOS ONE

Additional Editor Comments (optional):

Reviewers' comments:

Reviewer's Responses to Questions

**Comments to the Author**

1. If the authors have adequately addressed your comments raised in a previous round of review and you feel that this manuscript is now acceptable for publication, you may indicate that here to bypass the “Comments to the Author” section, enter your conflict of interest statement in the “Confidential to Editor” section, and submit your "Accept" recommendation.

Reviewer #1: All comments have been addressed

Reviewer #2: All comments have been addressed

2. Is the manuscript technically sound, and do the data support the conclusions?

Reviewer #1: Yes

Reviewer #2: Yes

3. Has the statistical analysis been performed appropriately and rigorously? 

Reviewer #1: Yes

Reviewer #2: Yes

4. Have the authors made all data underlying the findings in their manuscript fully available?

Reviewer #1: Yes

Reviewer #2: Yes

5. Is the manuscript presented in an intelligible fashion and written in standard English?

Reviewer #1: Yes

Reviewer #2: Yes

6. Review Comments to the Author

Reviewer #1: Accepted. No need more revision. All comments have been addressed. The authors have adequately addressed your comments raised in a previous round of review and you feel that this manuscript is now acceptable for publication

Reviewer #2: (No Response)

7. PLOS authors have the option to publish the peer review history of their article (what does this mean?). If published, this will include your full peer review and any attached files.

Reviewer #1: No

Reviewer #2: **Yes: **Dr. Iván Santolalla Arnedo

---

## [Editor Report · Acceptance letter]

28 May 2024

PONE-D-23-36317R1 

PLOS ONE

Dear Dr. Marcinowicz, 

I'm pleased to inform you that your manuscript has been deemed suitable for publication in PLOS ONE. Congratulations! Your manuscript is now being handed over to our production team.

Kind regards, 

on behalf of

Dr. Nabeel Al-Yateem 

Academic Editor

PLOS ONE